# Inverse Correlation of Th2-Specific Cytokines with Hepatic Egg Burden in *S. mansoni*-Infected Hamsters

**DOI:** 10.3390/cells13181579

**Published:** 2024-09-20

**Authors:** Lena Russ, Verena von Bülow, Sarah Wrobel, Frederik Stettler, Gabriele Schramm, Franco H. Falcone, Christoph G. Grevelding, Martin Roderfeld, Elke Roeb

**Affiliations:** 1Department of Gastroenterology, Justus Liebig University, 35392 Giessen, Germany; lena.russ@gmx.de (L.R.); verena.von-buelow@innere.med.uni-giessen.de (V.v.B.); martin.roderfeld@innere.med.uni-giessen.de (M.R.); 2Early Life Origins of Chronic Lung Diseases, Priority Research Area Chronic Lung Diseases, Research Center Borstel, Leibniz Lung Center, 23845 Borstel, Germany; 3Institute of Parasitology, BFS, Justus Liebig University, 35392 Giessen, Germany; franco.falcone@vetmed.uni-giessen.de (F.H.F.); christoph.grevelding@vetmed.uni-giessen.de (C.G.G.)

**Keywords:** *S. mansoni*, schistosomiasis, immune response, cytokines

## Abstract

Schistosomiasis, a parasitic disease caused by *Schistosoma* spp., affects more than 250 million people worldwide. *S. mansoni* in particular affects the gastrointestinal tract and, through its eggs, induces a Th2 immune response leading to granuloma formation. The relationship between egg load and immune response is poorly understood. We investigated whether the quantity of parasitic eggs influences the immune response in *S. mansoni*-infected hamsters. The hepatic and intestinal egg load was assessed, and cytokine expression as well as the expression of three major egg-derived proteins were analyzed in monosex- and bisex-infected animals by qRT-PCR. Statistical correlations between egg load or egg-derived factors *Ipse/alpha-1*, *kappa-5*, and *omega-1*, and the immune response were analyzed in liver and colon tissue. Surprisingly, no correlation of the Th1 cytokines with the hepatic egg load was observed, while the Th2 cytokines *Il4*, *Il5*, and *Il13* showed an inverse correlation in the liver but not in the colon. A longer embryogenesis of the parasitic eggs in the liver could explain this correlation. This conclusion is supported by the lack of any correlation with immune response in the colon, as the intestinal passage of the eggs is limited to a few days.

## 1. Introduction

Schistosomiasis is one of the most common parasitic diseases worldwide, along with malaria [1]. It is estimated that at least 250 million people worldwide are affected by this neglected tropical disease, although a high number of unreported cases must be assumed [2]. Schistosomiasis, also known as bilharziasis, belongs to the parasitic diseases caused by trematodes of the genus *Schistosoma* spp. [3]. The disease is very common in sub-Saharan Africa, the Middle East, South America, the Caribbean and Southeast Asia. According to WHO estimates, around 90% of those affected and in need of treatment live in Africa [4]. The reasons for the wide spread and transmission in endemic areas are warm climate, prevailing poverty, inadequate healthcare systems, insufficient sanitary facilities, and lack of access to clean drinking water [5]. As a result of climate change and globalization, tropical diseases are spreading to new areas [6]. It is assumed that schistosomiasis will spread to southern Europe in the coming years [7]. An outbreak in Corsica was reported as early as 2014, in which tourists and locals were infected [8]. Several species of Schistosomes are pathogenic to humans, the most common being *Schistosoma mansoni* (*S. mansoni*), *S. haematobium*, and *S. japonicum* [9].

The parasites use freshwater snails native to warm, standing waters as an intermediate host to produce cercariae, their larval stage, which is infectious to humans and other final vertebrate hosts. Cercariae penetrate the hosts’ skin upon contact, mature in the vascular system to adult worms, and finally settle in the mesenteric veins producing eggs. [9]. *S. mansoni* and *S. japonicum* predominantly infest the gastrointestinal tract, *S. haematobium* the urogenital tract [10]. In endemic areas, chronic schistosomiasis plays a particularly important role. Initially, schistosomiasis manifests through non-specific symptoms and can lead to dangerous long-term consequences [11]. The most serious complication is the development of hepatosplenic schistosomiasis (HSS), which is characterized by portal hypertension and its consequences [11].

Murine schistosomiasis has been employed as a model for studying various aspects of human schistosomiasis, especially the development of hepatic granulomatous inflammation with similar dynamics and cellular compositions [12]. In this context, the liver plays an important role in evading host immunity, as it is able to induce immune tolerance [13]. In the first weeks of a murine *S. mansoni* infection model, while the host was exposed to schistosomula and maturing schistosomes, a Th1-like immune response was observed [14]. However, with the onset of egg deposition, a pronounced Th2 immune response was induced and characterized by high production of interleukin-4 (IL-4) and IL-13 [15]. Furthermore, schistosomiasis is associated with squamous cell carcinoma of the urinary bladder (*S. haematobium*), and it has been discussed as a promoting factor for hepatocellular carcinoma (*S. mansoni*, *S. japonicum*) [16,17]. The only drug available for the treatment of schistosomiasis is Praziquantel, which is effective against adult worms exclusively. However, juvenile stages, intestinal eggs, and eggs trapped in the hepatic sinusoids are unaffected. The granulomatous fibrosis around the trapped eggs is mainly caused by cellular immune reactions and orchestrated by CD4^+^ T cells [18]. The modulation of T cells, through CTLA-4-IgG for instance, might improve hepatic fibrosis induced by *S. mansoni* eggs and constitutes a possible preventive approach [18]. The aim of this study was to investigate whether the extent of *S. mansoni*-induced hepatic and intestinal local cytokine expression is dependent on the quantity of parasite eggs or expression levels of the egg-derived proteins IPSE/alpha-1, kappa-5, and omega-1, which provoke a strong immunomodulatory capacity and have been found to be the primary targets of the egg-directed antibody response of the host [19].

## 2. Materials and Methods

### 2.1. Animal Model

*Biomphalaria glabrata* snails served as intermediate hosts, and female Syrian hamsters (*Mesocricetus auratus*) as final hosts for maintaining the life-cycle of a Liberian strain of *S. mansoni* [20]. Hamsters were infected with *S. mansoni* cercariae by the paddling method [21] at the age of 8 weeks. The golden hamsters were infected at the age of 8 weeks by bath infection with cercariae in water at room temperature (45 min, 1750 cercariae). Bs (bisex) and ms (monosex) worm populations were produced by poly-miracidial and mono-miracidial intermediate-host infections, respectively [21]. For the ms group, cercariae were initially isolated and then multiplied, so that infection with both sexes but only with one sex per hamster occurred. In total, 22 bs infections were carried out at 8 weeks of (hamster) age, and infected animals maintained for 46 days. In addition, 10 ms infections were carried out for 67 days to ensure a complete maturation of the worms [22]; females take longer to grow and develop more slowly in the absence of males. Six non-infected (ni) animals were used as super-controls. The livers of the animals were perfused in situ, removed, shock-frozen with liquid nitrogen and stored at −80 °C.

All animal experiments were performed in accordance with the European Convention for the Protection of Vertebrate Animals used for experimental and other scientific purposes (ETS No 123; revised Appendix A) and were approved by the Regional Council Giessen (V54-19 c 20/15 c GI 18/10 Nr. A26/2018).

### 2.2. Hematoxylin and Eosin Staining

Sections of liver lobes from bisex-infected hamsters were stained with hematoxylin and eosin (H&E) in total cross-section.

### 2.3. Quantitative Real-Time PCR

The isolation of mRNA was performed with the RNeasy^®^ Mini Kit from Qiagen (Cat. No. 74106) according to the manufacturer protocol. cDNA synthesis was performed with iScript cDNA Synthesis Kit from Bio-Rad (Cat. No. #1708891) (Bio-Rad Laboratories, Inc., Hercules, California, USA) according to manufacturer protocol. RT-qPCR was performed as described recently [23]. Intron-spanning primers were used (Primer list: Appendix A).

For qRT-PCR analysis, we used the 2^−ΔΔCT^ method [24]. The cycle in which the signal strength exceeds a defined value T (threshold) is referred to as the threshold cycle CT. The less starting DNA is present in the sample, the more cycles must be run through until the signal reaches the threshold value and the greater the CT value. With the CT value of each sample, the starting DNA can be relatively quantified using the 2^−ΔΔCT^ method according to Livak and Schmittgen [24]. The CT value of each sample is normalized to the average value of a reference group and expressed as a multiple of this. In our study, the non-infected cohort served as the reference group. It was only necessary to deviate from this when analyzing the *Il4* quantification. Here, the monosex-infected group was used as a reference, as the cDNA sought was not detectable in the non-infected group. For the quantification of the SEA component Ipse/α1, isolated *S. mansoni* eggs were used as the reference group. In order to compensate for possible fluctuations in the total DNA quantity of the samples, the expression of a reference gene was also normalized. The primers were produced by Microsynth and diluted to a concentration of 10 μM. The Platinum^®^ SYBR^®^ Green qPCR SuperMix-UDG (Invitrogen, Carlsbad CA, USA) was used as the reaction mix. This contains a Taq DNA polymerase, the dyes SYBR^®^ Green and ROX, reaction buffer (with Mg^2+^ ions), dNTPs and the enzyme uracil-DNA glycosylase (UDG). The dNTP mix contains, among other things, uracil as a base, which is incorporated into the DNA instead of thymine. In order to avoid carry-over contamination with PCR products from previous reactions, the UDG cleaves uracil from the DNA backbone before the PCR begins, thus preventing the amplification of these sequences. The filled tube strips were centrifuged and analyzed using the StepOnePlus^®^ Real-Time PCR System (Applied Biosystems, Foster City CA, USA). It served simultaneously as a thermal cycler and fluorescence photometer, created optimal temperatures for each reaction step and recorded the fluorescence intensity of the labelled dsDNA after each cycle.

Housekeeping genes that are not regulated and are expressed independently of external influences were used as reference genes. Data on the stability of housekeeping genes in hepatic tissue of healthy golden hamsters can only be found sporadically in the literature. Miao, Wang et al. found *Hprt1* (hypoxanthine phosphoribosyltransferase 1), *Actb* (b-actin), *Tubb* (β-tubulin) and *Rps18* (40S ribosomal protein S18) to be the most stable [25]. In a study of human liver samples in hepatitis C infection, *Gusb* (β-glucuronidase) and *Srsf4* (serine/arginine-rich splicing factor 4) were found to be suitable [26]. Even in severely inflamed and fibrosed tissue, their expression remained stable [26]. Based on these data, the six housekeeping genes mentioned above were reviewed for the present work. For each gene, qRT-PCR was performed, and the CT values of all samples (bisex *n* = 22, monosex *n* = 10, and non-infected *n* = 6) were analyzed for group differences. In addition, the web-based program RefFinder [27] was used to determine and compare the deviation of gene expression in the entire sample collective (*n* = 38). *Srsf4* showed the most stable expression. Regression analyses between egg load and *Srsf4* expression within the bisex-infected group also showed no correlation between the two parameters. 

### 2.4. Determination of the Hepatic and Colon Egg Load

Schistosome eggs were isolated by digestion of randomly chosen parts of liver and colon tissue (without feces) of bs-infected hamsters with potassium hydroxide (KOH) according to Cheever [28]. The even distribution of liver eggs is shown in Figure 1. One ml of 5% KOH solution was used per 100 mg of tissue. Approximately 100 mg of tissue was digested per animal. To accelerate tissue degradation, tissue digestion occurred at 37 °C for 16 h.

### 2.5. Statistical Analysis

Statistical analyses were performed with SPSS 26.0. (SPSS Inc., IBM Corp., Armonk, NY, USA). With regard to the exploratory nature of the current study, Kruskal–Wallis test without post hoc correction was used to calculate pairwise differences between animal groups [29]. Absolute values are presented, and statistic calculations were performed with transformed data. Data were skewed to the right and thus transformed using the natural logarithm. Relevant *p*-values are indicated in the corresponding figures.

## 3. Results

### 3.1. Quantification and Characterization of the Hepatic and Colon Egg Load

Hamsters were infected with *S. mansoni*, and the egg burden and expression of egg-specific antigens as well as the expression of characteristic cytokines were assessed (Figure 1A). The egg burden of bs-infected hamsters was determined after KOH digestion of liver and colon tissues (Appendix A). The egg load of the animals showed a wide range, from 12 to 85 eggs/mg liver tissue (mean 43.8, median 42.7, standard deviation 18.9) and 24–109 eggs/mg colon tissue (mean 57.3, median 54.0, standard deviation 24.5).

The liver sections were randomly selected to determine the hepatic egg load. To check whether the distribution of parasite eggs in the parenchyma of a liver lobe was uniform, HE staining was performed. Figure 1B shows HE staining of a representative section. In the overview image (Figure 1B, left panel), a complete liver lobe is shown, and eggs are marked in red. The distribution of the eggs is relatively even, comprising neither large egg-free areas nor large egg clusters. On closer inspection (Figure 1B, right), the eggs and their surrounding granulomas can be easily distinguished from intact liver parenchyma.

The eggs secrete various proteins into the surrounding tissue, the so-called ESPs (excretory/secretory proteins or egg-secreted proteins). With ~80%, the glycoprotein IPSE/alpha-1 represents the main component of liver ESPs [30]. In case the egg load of the tissue was high, we expected an increased IPSE/alpha-1 content. In order to check this, the potential correlation of egg load and *Ipse/alpha-1* expression was investigated. Quantification of *Ipse/α1* expression in the bs-infected group was performed using qRT-PCR. As expected, both control groups (ni or ms-infected without egg production) showed no *Ipse/α1* expression. The regression analysis showed a positive exponential correlation between *Ipse/α1* expression and egg load in the liver (Figure 1C, R^2^ = 0.862). As the assessment of *Ipse/α1* expression was performed by the 2^−ΔΔCt^-method [26] with different housekeeping genes as references, no absolute comparison of transcriptional levels of *Ipse/α1* in liver and colon was possible (Figure 1D). Notably, the varying degree of egg load in the colon correlated with the individual worm burden (Appendix A).

### 3.2. S. mansoni Infection-Induced Th1 Cytokine Response Is Independent of Hepatic Egg Burden 

At the beginning of infection with *S. mansoni*, the host usually reacts with a Th1 immune response. This reaction is typically characterized by the appearance of the cytokine IFN-γ. Therefore, we determined *Ifnγ* expression in all three experimental groups (Figure 2A). Compared to the ni control group, *Ifnγ* expression was significantly higher in the ms- and bs-infected groups (Figure 2A). While the median of both infection groups hardly differed (ms 49.5; bs 54.2), the values of the ms-infected group showed a larger scattering. *Ifnγ* expression of the bs-infected group and egg load showed no correlation (Figure 2B). Also, *Tnfα* expression was significantly higher in the ms- and bs-infected groups compared to the ni-control (Figure 2C). While the median of both infection groups hardly differed (ms 5.7; bs 5.0), the values of the ms-infected group showed a larger scattering. Similarly to *Ifnγ*, *Tnfα* expression in the bs-infected group showed no correlation between the *Tnfα* expression and egg load (Figure 2D). Notably, the induction of *Ifnγ* but not *Tnfα* in bs-infected animals was dependent on the egg burden in the colon (Figure 2E–H).

### 3.3. Th2 Cytokine Expression Inversely Correlated with the Hepatic Egg Burden

After the parasite starts producing eggs, the hepatic Th1 immune response shifts to a Th2 response [9], with IL-4, IL-5, and IL-13 being the prevalent cytokines involved. Their mRNA levels were determined in the liver samples of bs-, ms-, and ni-animals. *Il4*, *Il5*, and *Il13* showed different expression levels between the groups (Figure 3A,C,E). In the ni control group, the mRNAs of these three cytokines were hardly detectable. A slight cytokine expression was measured in the ms-infected group, whereas bs-infected animals showed up to 4.6-fold (*Il4*), 10-fold (*Il5*), and 30-fold (*Il13*) increased mRNA levels. Regression analyses showed an inverse linear dependence of cytokine expression on egg load (Figure 3B,D,F). The coefficients of R^2^ determination were 0.560 (*Il4*), 0.432 (*Il5*) and 0.438 (*Il13*), respectively.

### 3.4. Induction of Il4, Il5, and Il13 mRNA Expression in Colon Tissue Showed No Dependence on Intestinal Egg Burden

*Il4* expression was induced in the bowel of bs-infected hamsters (Figure 4A). However, we observed no correlation between the *Il4* expression and the intestinal egg burden (Figure 4B). Notably, *Il5* and *Il13* expression levels showed no group differences and, therefore, no correlations with the intestinal egg load (Figure 4C,D).

### 3.5. Hepatic Il4 and Il13 mRNA Expression Correlated Inversely with Egg-Derived Factors of S. mansoni Eggs

The Th2 response in *S. mansoni* infection is induced by egg-derived products such as glycans and glycoproteins, Omega-1 in particular [31]. However, IPSE/alpha-1 has not been shown to induce Th2 responses, although it triggers the release of IL-4 and IL-13 from basophils [15,19]. In liver tissue, we observed an inverse correlation between the expression of Th2 cytokines (*Il4* and *Il13*) and soluble egg antigens *Ipse/α1 and Omega1* and to a lower extent, also *Kappa5* (Figure 5).

Particularly high *Ipse/α1*, *Kappa5*, and *Omega1* levels were found to be associated with low *Il4* and *Il13* expression. Relative quantification was calculated from qRT-PCR data using the 2^−∆∆CT^ method and normalized to the ms-infected group or, in the case of *S. mansoni* genes, to an artificially induced control of cDNA derived from mRNA isolated from *S. mansoni* eggs. Mean values were calculated from three technical replicates each. Biological replicates covered bs-infected (*n* = 22) and ms-infected (*n* = 10) animals. The equal distribution of expression values between the groups was analyzed using the Kruskal–Wallis test. The dependence of the expression of *Il4* and *Il13* on the gene expression of egg-derived proteins was determined by regression analysis.

### 3.6. Expression of Il-10 in Liver and Colon Was Independent of the Egg Burden

In the chronic phase of infection, the Th2 response is controlled by anti-inflammatory measures including IL-10, Tregs, and alternatively activated macrophages (AAM) [15]. Accordingly, the *Il10* mRNA levels of bs-, ms- and ni-hamsters were quantified in liver and colon samples.

*Il10* was induced in the livers of ms- and bi-infected animals (Figure 6A). The variation within the ms-infected group was large, ranging from 3.9 to 17.5-fold compared to the ni group. The *Il10* expression within the bs-infected group was relatively constant, with a median of 10 times higher compared to the reference group. Regression analysis showed no correlation between *Il10* expression and egg load (Figure 6B).

*Il10* was induced in the colon of bs-infected animals (Figure 6C). Contrary to the observations in the liver, *Il10* expression was not induced in the colon of ms-infected hamsters. The variation within the bs-infected group was also large, ranging from 2 to 6.5-fold compared to the ni group. The *Il10* expression within the ms-infected group was relatively constant, within the range of the control group. Regression analysis showed no correlation between *Il10* expression and the intestinal egg load (Figure 6D).

## 4. Discussion

An infection with *S. mansoni* initially triggers a Th1 immune response, followed by a shift to a Th2 response shortly after egg production [32]. Additionally, egg production induces metabolic reprogramming and the development of reactive oxygen species (ROS) in hepatic tissue [33,34]. In this context, it has been shown that Omega-1 as ESP product is cytotoxic, whereas IPSE/alpha-1 is not [35]. Our study investigated the influence of egg quantity on the intensity of associated cytokine expression. To this end, we focused on the influence of egg burden on Th1 and Th2 immune responses in the liver and intestinal parenchyma. Cytokine expression from liver and colon tissue of bs-*S. mansoni*-infected hamsters, as well as soluble factors expressed by *S. mansoni* eggs [36], were quantified and correlated with the egg load in these tissues. To delineate the parenchymal effects induced by worms, samples from ms-infected hamsters were compared to ni-controls. As in pulmonary granuloma induced by *S. mansoni* eggs, IL-4 and IL-13 cooperate to initiate rapid Th2 cell–driven responses, and although their functions overlap, they perform additive roles [37]. Conversely, the decrease in the corresponding cytokines is associated with a decrease in the Th2 response.

Our data indicate an inverse organ-dependent correlation between egg load and also soluble factors of parasitic eggs on the one hand and the Th2 immune response on the other hand. This inverse correlation was not observed in the colon tissue of infected hamsters. Hepatic Th1 response did not correlate with hepatic egg load. Vital eggs secrete proteins, with the glycoprotein IPSE/alpha-1 as quantitatively dominating factor. Our analysis of different samples from one liver indicated a positive correlation with a high coefficient of determination between egg load and *Ipse/α1* expression (R = 0.862).

### 4.1. Egg Load-Independent Th1 Immune Response

The host organism initially reacts to an *S. mansoni* infection with an immune response characterized by the appearance of Th1 lymphocytes and cytokines such as IFN-γ [10]. In this study, infected animals showed a higher hepatic *Ifnγ* expression compared to the healthy control group. The Th1 immune response is triggered by worms and larvae even before parasitic eggs are laid [10]. In an infection situation, especially patients in endemic regions may harbor simultaneously eggs, adult worm pairs, unpaired adults, juvenile worms, and eventually cercariae during their skin passage. Therefore, an increase in *Ifnγ* expression might depend on different factors, including the time point of measurement. In our study, we can neglect a potential influence of juvenile worms and cercariae since the used hamsters were infected only once, and the time point of organ analyses was set when all parasites had fully developed. Previous studies indicated that egg components such as LNFPIII can induce IFN-γ production by natural killer cells and CD4+ T lymphocytes [38]. These T lymphocytes play a role in the formation of early granulomas. However, the *Ifnγ* values in animals of the bs-infected group showed no dependence on the egg load, refuting the hypothesis that the eggs are a significant trigger of the Th1 response.

### 4.2. Induction of the Th2 Immune Response in Bs-Infection in S. mansoni

During the time course of infection, the Th1 immune response alters to a Th2 response, with increased levels of IL-4, IL-5, and IL-13 [10]. The expression of these cytokines was particularly high in the bs-infected group. The eggs are responsible for the transition to a Th2 response, and also, soluble egg antigens (SEA) stimulate IL-4 and IL-13 production [39]. The ms-infected group also showed a moderate increase in cytokine expression, which is most likely caused by an early Th2 response triggered by adult worm antigens [40]. Both, worms and eggs possess cross-reactive antigens that enable a rapid recruitment of CD4+ T lymphocytes after egg deposition [40]. Furthermore, there are indications that host sensitization through worm antigens is absolutely necessary in order to be able to form granulomas after egg deposition [41]. Even larval precursors appear to be able to promote the maturation of CD4+ T lymphocytes to some extent and increase Th2 cytokine levels [42].

### 4.3. Reduction of the Th2 Immune Response with Increasing Egg Load

A closer examination of the bs-infected group revealed a significant reduction in *Il4*, *Il5*, and *Il13* expression with increasing egg load. Although IL-4 and IL-13 were considered crucial for granuloma formation, a combined depletion of *Il4* and *Il13* in infected mice impaired egg excretion, leading to endotoxaemia and acute mortality [43]. This indicated that Th2 cytokines are involved in inducing regulated anti-inflammatory responses, in particular the induction of AAM [44]. 

In our analysis, liver samples with particularly high *Ipse/α1* levels showed very low *Il4* expression. To our knowledge, the correlation between a decrease in *Il4*, *Il5*, and *Il13*—and as such, probably the Th2 immune response—with increasing egg burden has not been described before. Therefore, we speculate that the host may have developed a strategy to protect itself from the repercussions of an excessive immune reaction by limiting the Th2 response in the case of high egg load. As the liver is regularly exposed to various pathogens and antigens from portal vein blood, it generally has a high immune tolerance, thus avoiding hypersensitive reactions [45]. Consequently, a reduction in the Th2 response has also been observed in the chronic stage of schistosomiasis to prevent tissue damage [46]. This results in the development of hypo-responsive T lymphocytes and even T-cell anergy and decreasing Th2 cytokine levels [45]. Recently, Peterkova et al. observed that *S. mansoni* eggs differ substantially based on their tissue localization. Their robust comparative transcriptomic analysis showed differences between intestinal and liver-entrapped eggs. Genes associated with development and metabolism, as well as known immunomodulators, show major differences in gene expression comparable to our analysis. The authors suspect a differential capacity of eggs to elicit a host immune response [47]. Notably, we observed gene expression of Ipse/α1 in liver and colon, while Peterkova et al. described that IPSE/alpha-1 and omega-1, together with micro-exon genes, are predominantly expressed in liver eggs’. Correspondingly, epidemiological studies showed a reciprocal relationship between the occurrence of chronic schistosomiasis and type I allergies [48,49].

The development of immune tolerance can also be observed in other chronic liver infections. Chronic hepatitis B and C as well as malaria were also shown to be associated with a reduced immune response. T lymphocytes are exhausted, particularly when the antigen load is high [50]. SEA can induce the apoptosis of Th cells in vitro by expressing Fas ligand, eliminating about 20% of CD4+ cells in this manner [51].

Remarkably, the inverse correlation of Th2-specific cytokines with egg burden observed in the liver was not observed in the colon, which suggests organ-dependent differences in immunological reactions and immune tolerance in a schistosomiasis background. Eggs in the gastrodermis are programmed to exit the host via tissue passage and gut lumen transport. In contrast, eggs indefinitely accumulate in the liver, which represents a deadlock for the parasite. These discrepancies of our results with regard to both tissues may result from the prolonged phase of the eggs in the liver [52], which may result in full egg development in this organ, while the intestinal passage of the eggs is limited to a few days, probably resulting in fewer fully developed eggs. As one of the consequences of this temporary organ presence, the accumulation of IPSE/alpha-1 and other ESPs in the gastrodermis/gut may be less pronounced compared to the liver.

Although the relative transcriptional expression levels in the liver and colon are not directly comparable, the exponential trend of the correlation of *Ipse/α1* expression and egg load is definitely steeper in the liver than in the gut. The steeper trend of the increase in *Ipse/α1* expression with the egg load in the liver might be an explanation for a stronger hepatic immune tolerance than in the colon.

There are some limitations of our study. The data presented were solely obtained from female animals. The tissue analyzed was obtained from the animals during the acute stage of schistosomiasis, 6.5 weeks after infection. Assessment of further time points after infection would give further insight into time kinetics and allow for more sophisticated conclusions. Many different molecules are involved in the immune response, metabolic pathways, and antioxidant reactions. In this study, individual cytokines representing Th1 and Th2 response were evaluated as representative molecules. The analysis of further factors is needed for a more detailed synopsis, but which goes far beyond this study. Finally, the lack of a quantitative expression pattern for IPSE/α1 was obvious. However, the development of a specific ELISA for IPSE/α1 was not possible within the framework of the current project.

We were able to show that *S. mansoni* eggs are evenly distributed in the liver of infected hosts. However, the individual hepatic egg load greatly varied between animals, although all animals were infected according to one protocol. The “paddling method” allows for little control over the number of cercariae infecting each individual animal, since the infection efficiency depends on the number of cercariae successfully penetrating the host skin and the ratio of male and female cercariae. This discrepancy in the resulting individual hepatic egg loads directly impacts the expression levels of egg-derived proteins, which in turn may be the trigger for the activation of the immune response or immune tolerance. Notably, the extent of hepatic accumulation of *S. mansoni* eggs inversely influenced the hepatic Th2 immune response, which might have been a consequence of extended Th2 cell exhaustion in the liver. This observed inverse correlation was absent in intestinal tissue. The very complex interplay of different cell types, antigens, cytokines, and other molecules requires further investigation, ideally including appropriate knockout animal models and the use of primary liver cells.

## Figures and Tables

**Figure 1 cells-13-01579-f001:**
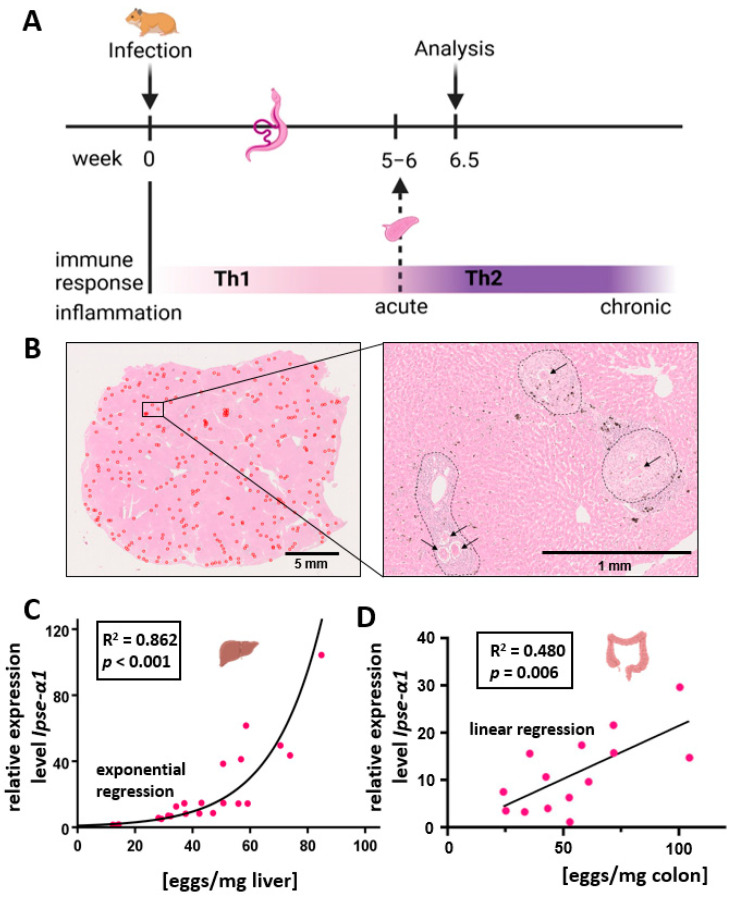
*S. mansoni* infection caused individual hepatic and colon egg burdens with different patterns of *Ipse/α1* expression. (**A**) Experimental setting. Created in BioRender. Roderfeld, M. (2023) BioRender.com/o55m552. (**B**) Uniform distribution of eggs within a liver lobe. The liver lobes of bs-infected hamsters were stained with HE in total cross-section. Overview of a liver lobe in cross-section with parasitic eggs (red circles). Enlargement with eggs (→) and granulomas delineated by (---). (**C**,**D**) Regression analysis demonstrated that the correlation of egg load and *Ipse/α1* fitted best with an exponential curve in the liver and a linear dependence in the colon. The expression of hepatic and colon *Ipse/α1* was analyzed by qRT-PCR in the bisex group and evaluated using the 2^−∆∆CT^ method (liver *n* = 22, colon *n* = 14). Three technical replicates were performed for each tissue. Relative quantification was calculated from qRT-PCR data using the 2^−∆∆CT^ method [24] and normalized to the monosex-infected group or, in the case of *S. mansoni* genes, to an artificially introduced control of cDNA derived from mRNA isolated from *S. mansoni eggs*.

**Figure 2 cells-13-01579-f002:**
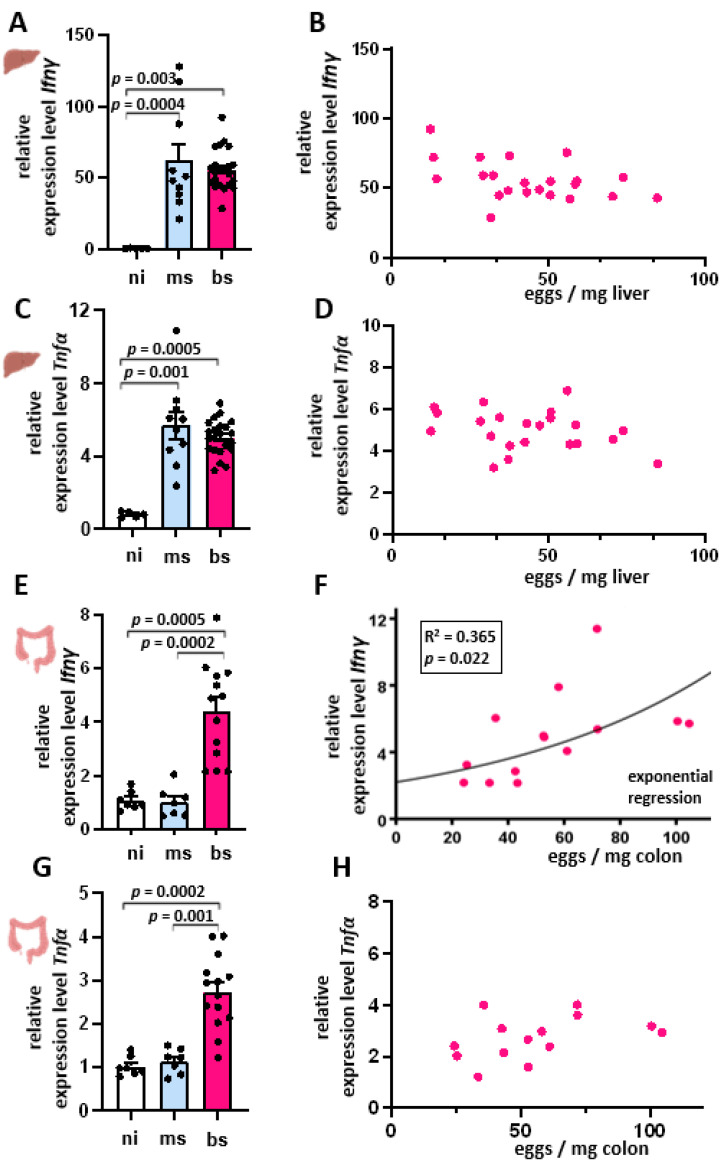
(**A**–**D**) *S. mansoni* infection-induced Th1 cytokine response was independent of hepatic egg burden. (**A**) Increased *Ifnγ* expression in livers of ms- (*n* = 10) and bs-*S. mansoni*-infected hamsters (*n* = 22). (**B**) No correlation between *Ifnγ* expression and hepatic egg load in bs-infected hamsters was observed. (**C**) *Tnfα* expression increased in livers of ms- (*n* = 10) and bs-*S. mansoni*-infected hamsters (*n* = 22). (**D**) No correlation was found between *Tnfα* expression and hepatic egg load in bs-infected hamsters. After performing qRT-PCR, the analysis of the data was performed using the 2^−∆∆CT^ method. (**E**) *Ifnγ* expression was induced in the colon of bisex-infected hamsters. (**F**) *Ifnγ* expression in the colon exponentially correlated to the egg load in the colon. (**G**) *Tnfα* expression was induced in the colon of bisex-infected hamsters. (**H**) *Tnfα* expression showed no correlation to egg burden in the colon. The ni group (*n* = 6) served as control. Three technical replicates were performed, and their mean values were determined. The equal distribution of expression values between the groups was tested using the Kruskal–Wallis test (*p* < 0.001) (**A**,**C**,**E**,**G**). In the regression analyses, egg load served as the independent variable, and fold increase in *Ifnγ* mRNA as the dependent variable.

**Figure 3 cells-13-01579-f003:**
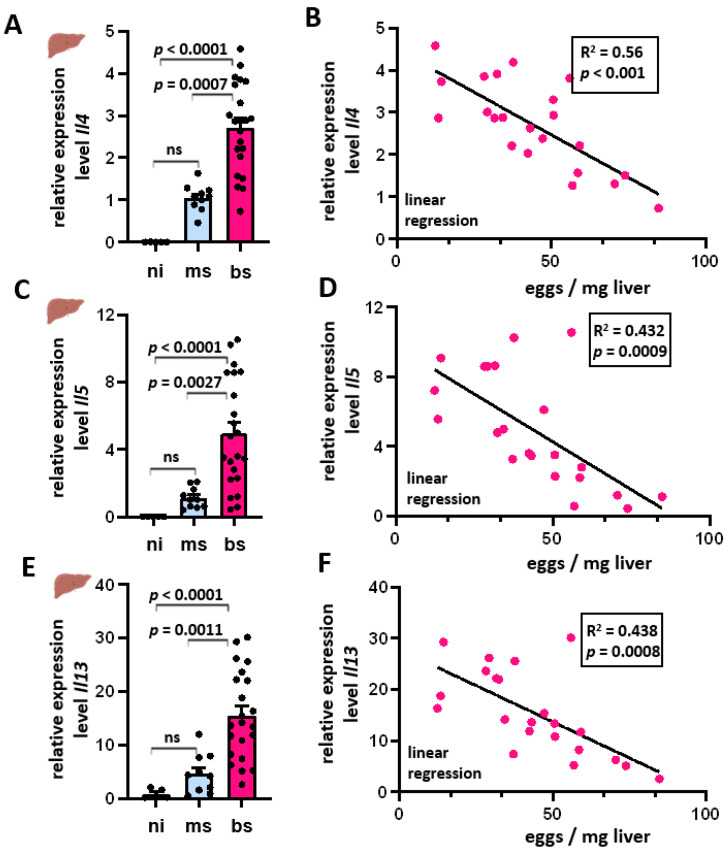
Inverse egg load-dependence of the hepatic Th2 cytokines IL-4, IL-5, and IL-13. (**A**,**C**,**E**) Increased expression (transcript) levels of *Il4* (**A**), *Il5* (**C**), and *Il13* (**E**) in the liver of infected hamsters. (**B**,**D**,**F**) Negative correlation between *Il4* (**B**), *Il5* (**D**), and *Il13* mRNA (**F**) and the respective egg load. Relative quantification was calculated from qRT-PCR data using the 2^−∆∆CT^ method and normalized to the ms- (**A**–**D**) or ni-groups (**E**,**F**). Mean values were calculated from three technical replicates each; biological replicates covered bs-infected (*n* = 22), ms-infected (*n* = 10), and ni (*n* = 6). The equal distribution of expression values between the groups was tested using the Kruskal–Wallis test ((**A**) *p* < 0.001; (**C**) *p* < 0.001). The linear correlation was determined by linear regression analyses.

**Figure 4 cells-13-01579-f004:**
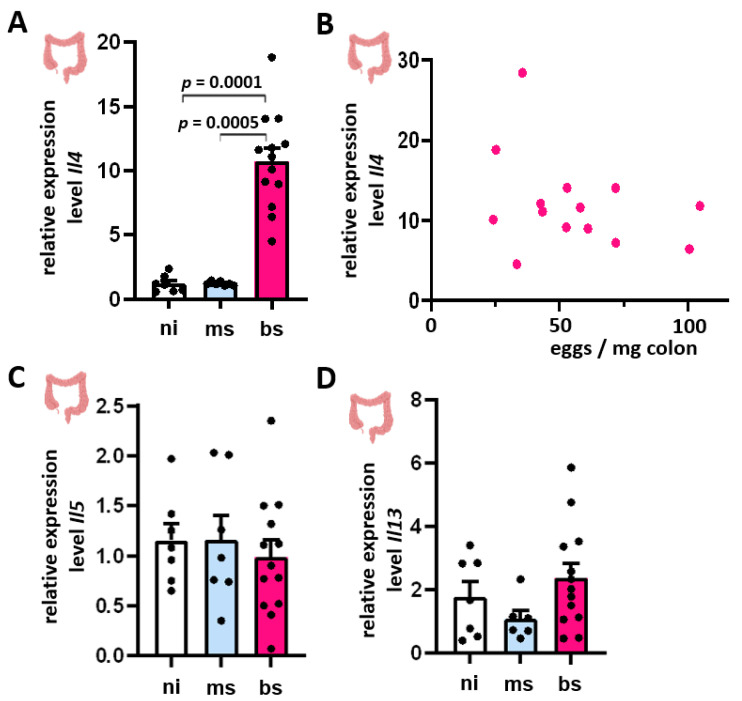
*S. mansoni* infection-induced *Il4* expression was independent of intestinal egg burden. (**A**,**C**,**D**) An increased expression of *Il4* (**A**), but not of *Il5* (**C**) and *Il13* (**D**), was observed in the bowel of infected hamsters. (**B**) No correlation was found between *Il4* and the respective intestinal egg load. The relative quantification was calculated from qRT-PCR data using the 2^−∆∆CT^ method and normalized to the ms-infected group (**A**,**C**,**D**). Mean values were calculated from three technical replicates each; biological replicates covered bs-infected (*n* = 14), ms-infected (*n* = 7), and ni (*n* = 7) animals. The equal distribution of expression values between the groups was analyzed using the Kruskal–Wallis test ((**A**) *p* < 0.001; (**C**) *p* < 0.001).

**Figure 5 cells-13-01579-f005:**
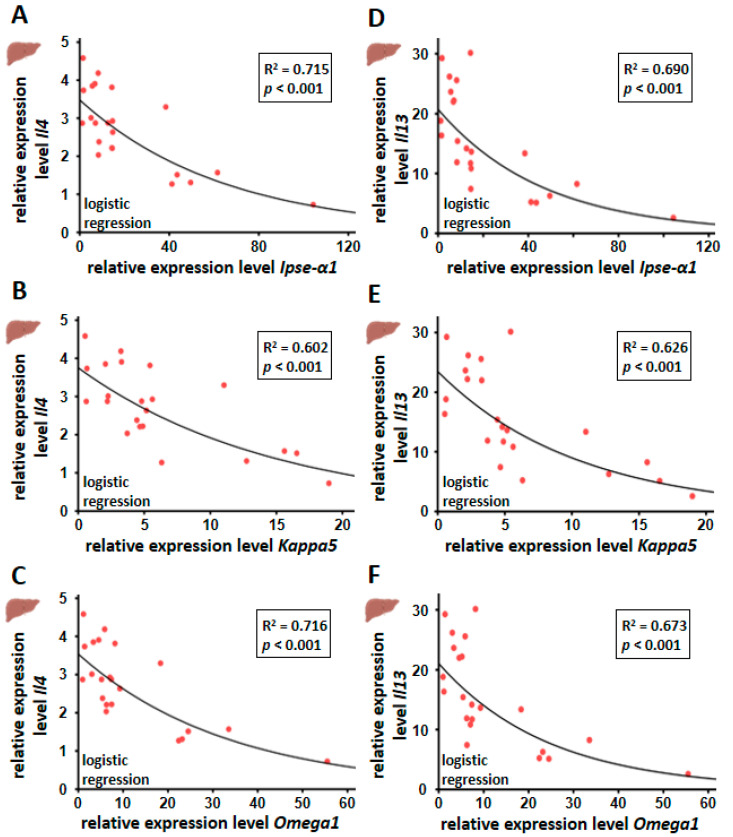
Inverse correlation between hepatic *Il4* (**A**–**C**) or *Il13* mRNA (**D**–**F**) and the gene expression of the different soluble egg-dependent factors *Ipse/α1*, *Kappa5* and *Omega1*.

**Figure 6 cells-13-01579-f006:**
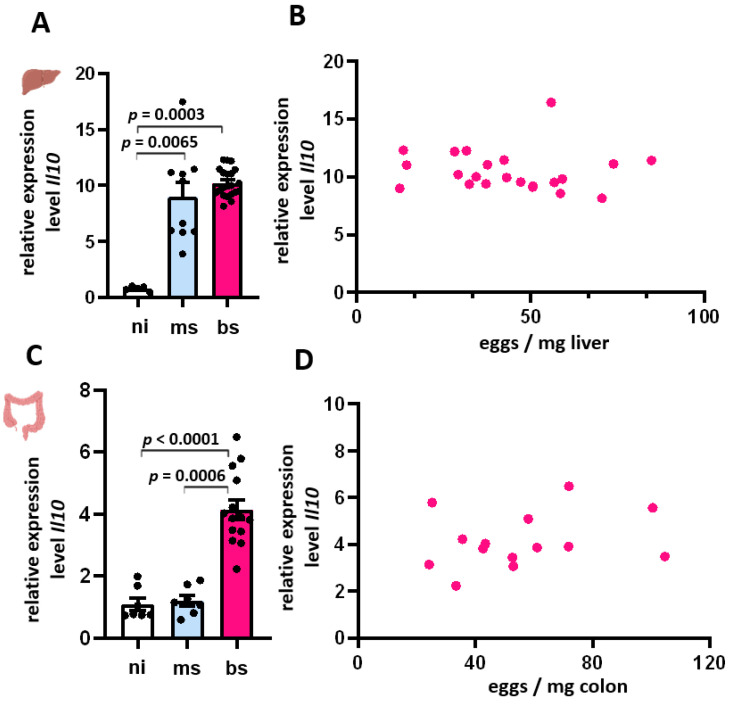
Expression of *Il10* in liver and colon was independent of the egg burden. (**A**) An increased expression of *Il10* in livers of ms- and bs-infected hamsters was observed. (**B**) We found no correlation between *Il10* and the hepatic egg load. (**C**) In the colon of bs-infected hamsters, we found an increased expression of *Il10*. (**D**) Also in the colon, we found no correlation between *Il10* and egg load. (**A**–**D**) mRNA quantification was performed by qRT-PCR and the 2^−∆∆CT^ method (normalization to non-infected group). Liver samples were obtained from bs-infected (*n* = 22), ms-infected (*n* = 10), and ni (*n* = 6) animals. Colon samples were obtained from bs-infected (*n* = 14), ms-infected (*n* = 7), and ni (*n* = 7) animals. Mean values were calculated from three replicates. The equal distribution of expression values between the groups was analyzed using the Kruskal–Wallis test ((**A**) *p* < 0.001). In the regression analysis, egg load served as the independent variable, and *Il10* as the dependent variable (**B**).

## Data Availability

The data that support the findings of this study are available from the corresponding author upon reasonable request.

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
