# Peer review of "Inverse Correlation of Th2-Specific Cytokines with Hepatic Egg Burden in S. mansoni-Infected Hamsters"

_cells, 2024, doi:10.3390/cells13181579_

Round 1
Reviewer 1 Report
Comments and Suggestions for Authors
I was pleased to read the manuscript. Russ et al. present a small-scale descriptive hamster study exploring the association between tissue cytokine expression and the number of Schistosoma mansoni eggs present in the liver and colon. The cytokine expression data are also correlated to the expression of major S. mansoni immunomodulatory factors.
Although lacking mechanistic insights, the study addresses an important biological question with possible implications for host-schistosome interactions. While some results are negative (showing no differences or correlations), the part demonstrating an inverse correlation between the Th2-promoting factors (especially Th2-driving omega-1) and Th2 cytokines is particularly interesting and should be further deciphered.
MAJOR ISSUES
(1) qRT-PCR is a central method of the study. Therefore, it should be stated already in M&M (a) how the qRT-PCR data were analyzed (ddCt is mentioned later in the text), (b) how the expression was normalized (to ni or ms group?), and (c) which genes were used as housekeeping/reference controls (both in mice and schistosomes). Reference [26] (line 101) does not contain sufficient details on the methodology. Moreover, why did the authors use different housekeeping genes (lines 161-162) for liver and intestinal samples? The comparison of the tissues (Fig. 1C, D) might be very insightful.
(2) The authors present data on the cytokine expression in liver and colon tissues. However, it might be helpful to see whether there is some "egg-dose" effect also on the antigen-specific T cell response (as a likely major source of the cytokines). In other words, the authors’ conclusions could be supported by experiments showing cytokine production by, e.g., mesenteric lymph node cells or splenocytes (originating from mice with different egg loads) stimulated by, e.g., soluble egg antigens. This would provide functional support, even on the protein (not only transcriptional) level.
(3) If I got it correctly, the authors speculate (lines 341-384) that the reduction of Th2-related cytokine expression in hamsters with higher egg load could be linked to a certain regulatory loop protecting the animals from excessive Th2 response (and pathology). This sounds reasonable, but I lack any evidence to support this. One could expect that IL-10 would play a role in this, which seems to be implicitly supported by discussion statements on “chronic stage,” “energy,” or “hyporesponsiveness,” which are in schistosomosis connected to the production of IL-10 (lines 346-348). However, and importantly, there are no correlations between Il10 expression and egg load, neither in the liver nor colon (Fig. 6B & 6D). In my view, it somewhat fits the infection stage (6.5 weeks) but contradicts the authors’ hypothesis.
(4) The authors present differences between liver and intestinal eggs several times (e.g., Fig. 1C & 1D, Fig. 2B and S2B, lines 364-366 & 374-377). Unfortunately, their expression data are not directly comparable (the scale is actually massively different). Have the authors checked the recent paper on differential transcriptional profiling of liver and intestinal eggs (Peterkova et al. 2024 - https://doi.org/10.1371/journal.ppat.1012268)? It might provide food for thought and allow meaningful comparisons/discussion.
(5) It turns out that S. mansoni strains differ in virulence and pathology/disease outcomes (Jutzeler et al. 2024 - https://doi.org/10.1186/s13071-024-06286-6). Could the authors discuss how their results obtained from the Liberian strain are possibly relevant for other S. mansoni strains, e.g. the more virulent (and prevalent in the laboratories) PR strain?
MINOR ISSUES
(a) The introduction should be divided into paragraphs for better orientation.
(b) Lines 53-54: "In this context, the liver plays an important role in evading host immunity as it is able to induce immune tolerance [12]." – The sentence seems a bit general and out of context.
(c) Lines 61-62: "In addition, co-infections with the human immunodeficiency virus (HIV), hepatitis B virus (HBV) or hepatitis C virus (HCV) are common [16,17]." – The sentence seems a bit out of context.
(d) Lines 72-73: I would specify the "immune response" as the authors analyzed only the local expression of certain cytokines.
(e) Lines 74-75: I suggest rephrasing the reason for choosing IPSE/alpha-1, kappa-5, and omega-1. Although antibodies target them, this is not much relevant to the cytokine-oriented study. I assume the immunomodulatory capacity of those molecules plays the main role.
(f) Line 80: Please, indicate the infection dose.
(g) Line 81: It is unclear if the monosex (ms) infection was performed with male or female cercariae.
(h) Line 86: How many non-infected (ni) animals were used?
(i) Line 84: It should be indicated that the hamsters were perfused (which I expect as the worm burden was also assessed in Fig. S1D).
(j) Lines 92-95: It is unclear what kind of histochemical staining/detection was performed. Please, specify. It seems that only HE (no histochemistry) was used.
(k) Line 112: Please, specify which part of the liver and colon was used for the analysis. Was the colon washed from the feces prior to weighting?
(l) Lines 117-123: I miss information on the regression/correlation analyses, which seem critical for most outcomes.
(m) Figure 5: I would invert the axis, putting the molecules on X axis ("cytokine expression is dependant on the molecule expression").
(n) Lines 282-285: The sentence is very long and difficult to understand.
(o) Lines 274-300: This part is quite long but does not discuss anything. I believe it is largely a summary that could be condensed so that the reader can focus on the discussion itself.
(p) Lines 383-384: "In this study, individual enzymes, cytokines, and transcription factors were evaluated as representative molecules." – I do not understand, which enzymes and transcription factors are addressed.
Comments on the Quality of English LanguageThe language is fine and understandable. Minor suggestions can be found above.
Author Response
Answers to reviewer 1
We would like to thank Reviewer 1 for the favorable summary of our work.
MAJOR ISSUES
(1) qRT-PCR is a central method of the study. Therefore, it should be stated already in M&M (a) how the qRT-PCR data were analyzed (ddCt is mentioned later in the text), (b) how the expression was normalized (to ni or ms group?), and (c) which genes were used as housekeeping/reference controls (both in mice and schistosomes). Reference [26] (line 101) does not contain sufficient details on the methodology. Moreover, why did the authors use different housekeeping genes (lines 161-162) for liver and intestinal samples? The comparison of the tissues (Fig. 1C, D) might be very insightful.
Authors´ response: We apologize for the missing method description and have now integrated the RT-PCR in detail (a), the type and method of normalization (b) and the choice of reference genes (c) into the method section of the revised manuscript. The following paragraph has been included:
“For qRT-PCR analyzation we used the 2-ΔΔCT method [26]. The cycle in which the signal strength exceeds a defined value T (threshold) is referred to as the threshold cycle CT. The less starting DNA is present in the sample, the more cycles must be run through until the signal reaches the threshold value and the greater the CT value. With the CT value of each sample, the starting DNA can be relatively quantified using the 2-ΔΔCT method according to Livak and Schmittgen [Livak 2001]. The CT value of each sample is normalized to the average value of a reference group and expressed as a multiple of this. In our study, the non-infected cohort served as the reference group. It was only necessary to deviate from this when analysing the Il4 quantification. Here, the monosex infected group was used as a reference, as the cDNA sought was not detectable in the non-infected group. For the quantification of the SEA component Ipse/α1, isolated S. mansoni eggs were used as the reference group. In order to compensate for possible fluctuations in the total DNA quantity of the samples, the expression of a reference gene was also normalized. The primers were produced by Microsynth and diluted to a concentration of 10 μM. The Platinum® SYBR® Green qPCR SuperMix-UDG (Invitrogen) was used as the reaction mix. This contains a Taq DNA polymerase, the dyes SYBR® Green and ROX, reaction buffer (with Mg2+ ions), dNTPs and the enzyme uracil-DNA glycosylase (UDG). The dNTP mix contains, among other things, uracil as a base, which is incorporated into the DNA instead of thymine. In order to avoid carry-over contamination with PCR products from previous reactions, the UDG cleaves uracil from the DNA backbone before the PCR begins, thus preventing the amplification of these sequences. The filled tube strips were centrifuged and analyzed using the StepOnePlus® Real-Time PCR System (Applied Biosystems). It served simultaneously as a thermal cycler and fluorescence photometer, created optimal temperatures for each reaction step and recorded the fluorescence intensity of the labelled dsDNA after each cycle.
Housekeeping genes that are not regulated and are expressed independently of external influences were used as reference genes. Data on the stability of housekeeping genes in hepatic tissue of healthy golden hamsters can only be found sporadically in the literature. Miao, Wang et al. found Hprt1 (hypoxanthine phosphoribosyltransferase 1), Actb (b-actin), Tubb (β-tubulin) and Rps18 (40S ribosomal protein S18) to be the most stable [Miao 2020]. In a study of human liver samples in hepatitis C infection, Gusb (β-glucuronidase) and Srsf4 (serine/arginine-rich splicing factor 4) were found to be suitable [Congiu 2011]. Even in severely inflamed and fibrosed tissue, their expression remained stable [Congiu 2011]. Based on these data, the six housekeeping genes mentioned above were reviewed for the present work. For each gene, qRT-PCR was performed and the CT values of all samples (bisex n = 22, monosex n = 10 and non-infected n = 6) were analyzed for group differences. In addition, the web-based programme RefFinder [Xie 2023] was used to determine and compare the deviation of gene expression in the entire sample collective (n = 38). Srsf4 showed the most stable expression. Regression analyses between egg load and Srsf4 expression within the bisex-infected group also showed no correlation between the two parameters.”
The following references have been included into the revised manuscript:
Miao J, Wang J, Robl N, Guo H, Song S, Peng Y, Wang Y, Huang S, Li X. 2020. Validation of Stable Housekeeping Genes for Quantitative Real Time PCR in Golden Syrian Hamster. Indian Journal of Animal Research(Of). doi:10.18805/IJAR.B-1237.
Congiu M, Slavin JL, Desmond PV. 2011. Expression of common housekeeping genes is affected by disease in human hepatitis C virus-infected liver. Liver international official journal of the International Association for the Study of the Liver 31(3):386–90. doi:10.1111/j.1478-3231.2010.02374.x.
Xie F, Wang J, Zhang B. 2023. RefFinder: a web-based tool for comprehensively analyzing and identifying reference genes. Functional & integrative genomics 23(2):125. doi:10.1007/s10142-023-01055-7.
(2) The authors present data on the cytokine expression in liver and colon tissues. However, it might be helpful to see whether there is some "egg-dose" effect also on the antigen-specific T cell response (as a likely major source of the cytokines). In other words, the authors’ conclusions could be supported by experiments showing cytokine production by, e.g., mesenteric lymph node cells or splenocytes (originating from mice with different egg loads) stimulated by, e.g., soluble egg antigens. This would provide functional support, even on the protein (not only transcriptional) level.
Authors´ response: No mesenteric lymph nodes were isolated from the hamsters used for these experiments. Lymph node extirpation would require specific surgical methods that are not available in our laboratory. In addition, a rapid surgical method is required to analyze the liver tissue, which contradicts mesenteric lymph node extirpation from the retroperitoneum. In another mouse project from our laboratory, we analyzed spleen size and weight in relation to the hepatic egg load. However, there are considerable differences in the age of the animals. And furthermore, only 36% of mice with infestation of the whole liver showed any eggs at all in the spleen. Spleen weight and spleen-to-body weight ratio were increased in infected mice and reduced in elder infected animals. Lymphatic hyperplasia and extramedullar hematopoiesis increased in the spleen of elderly mice infected with S. mansoni. To summarize, the stimulation of splenocytes from mice with different egg exposure to soluble egg antigens would first require the establishment of a suitable specific mouse model, which is not yet available.
(3) If I got it correctly, the authors speculate (lines 341-384) that the reduction of Th2-related cytokine expression in hamsters with higher egg load could be linked to a certain regulatory loop protecting the animals from excessive Th2 response (and pathology). This sounds reasonable, but I lack any evidence to support this. One could expect that IL-10 would play a role in this, which seems to be implicitly supported by discussion statements on “chronic stage,” “energy,” or “hyporesponsiveness,” which are in schistosomosis connected to the production of IL-10 (lines 346-348). However, and importantly, there are no correlations between Il10 expression and egg load, neither in the liver nor colon (Fig. 6B & 6D). In my view, it somewhat fits the infection stage (6.5 weeks) but contradicts the authors’ hypothesis.
Authors´ response: The cytokine IL-10 is an important modulator in the inflammatory process. It can suppress both the TH1 and TH2 response and prevents an excessive immune response. IL-10 plays a particularly important role in the chronic stage. The hepatic mRNA of bisex, monosex and non-infected hamsters was analyzed for Il10 mRNA. Both the livers of monosex and bisex infected animals show elevated levels compared to the super control. The variation within the monosex infected group was large, ranging from 3.9- to 17.5-fold compared to the uninfected group. The Il10 expression within the bisex infected group was relatively constant, with a median of 10 times that of the reference group. However, a regression analysis showed no correlation between Il10 expression and egg load. The lack of correlation in our model is probably due to the age of the hamsters, in which Th1 and Th2 response can still overlap as shown in figure 1A. During the course of infection, the TH1 immune response as cited in doi:10.3389/fimmu.2018.02492 transitions into a TH2 response with an increase in IL-4, IL-5, and IL-13 levels. The expression of these cytokines is strongly increased in our experimental subjects, especially in the bisex infected groups. This observation is consistent with many other studies. The eggs are responsible for the transition to a TH2 response and SEA stimulates IL-4 and IL-13 production. Due to its inhibitory effect on macrophage function, IL-4 - like interleukin-10 - is also categorized as an anti-inflammatory cytokine. In pulmonary granuloma IL-4 and IL-13 cooperate to initiate rapid Th2 cell–driven responses, and although their functions overlap, they perform additive roles [https://doi.org/10.1084/jem.189.10.1565].
Hepatic egg load does not appear to significantly influence Il-10 expression in this model. And the reviewer is right, there is no direct indication that IL-10 is responsible for the suppression of the TH2 cytokines IL-4 and IL-13 observed here. From mice of different age groups, however, we know that among all analyzed cytokines Il-10 expression showed the best correlation with the hepatic egg load.
The following sentences were included into the discussion to clarify this issue in line 348(original) and 507 of the revised manuscript:
“As in pulmonary granuloma induced by Schistosoma mansoni eggs, IL-4 and IL-13 cooperate to initiate rapid Th2 cell–driven responses, and although their functions overlap, they perform additive roles [McKenzie GJ JEM 1999]. Conversely, the decrease in the corresponding cytokines is associated with a decrease in the Th2 response.”
(4) The authors present differences between liver and intestinal eggs several times (e.g., Fig. 1C & 1D, Fig. 2B and S2B, lines 364-366 & 374-377). Unfortunately, their expression data are not directly comparable (the scale is actually massively different). Have the authors checked the recent paper on differential transcriptional profiling of liver and intestinal eggs (Peterkova et al. 2024 - https://doi.org/10.1371/journal.ppat.1012268)? It might provide food for thought and allow meaningful comparisons/discussion.
Authors´ response: We thank the reviewer for this valuable information. In fact, the expression data between liver and intestine are hardly comparable in quantitative terms, as the assessment of Ipse/α1 expression was done by the 2-ΔΔCt-method with different housekeeping genes as references, no absolute comparison of transcriptional levels of Ipse/α1 in liver and colon was possible (the same statement is in the results part describing Fig. 1 C and D). Nevertheless, in order to avoid misunderstandings from the massively different scales, we adapted the relative scale of Fig. 1D appropriately. The robust comparative transcriptomic analysis by Peterkova et al. also showed differences between intestinal and liver-entrapped eggs comparable to our analysis. Tissue localization affected gene expression of the host immune response depending on the specific antigenicity of the eggs in both tissues. These findings might indicate a significant two-way communication between the egg and the host. The following sentences were added to the line 377 of the original manuscript, and line 566 of the revised form.
“Recently Peterkova et al. oberserved that S. mansoni eggs differ substantially based on their tissue localization. Their robust comparative transcriptomic analysis showed differences between intestinal and liver-entrapped eggs. Genes associated with development and metabolism, as well as known immunomodulators, show major differences in gene expression comparable to our analysis. The authors suspect a differential capacity of eggs to elicit a host immune response [Peterková K, Konečný L, Macháček T, Jedličková L, Winkelmann F, Sombetzki M, et al. (2024) Winners vs. losers: Schistosoma mansoni intestinal and liver eggs exhibit striking differences in gene expression and immunogenicity. PLoS Pathog 20(5): e1012268]. Notably, we observed gene expression of Ipse/α1 in liver and colon while Peterkova et al. described that ´ IPSE/alpha-1 and omega-1, together with micro-exon genes, are predominantly expressed in liver eggs´.“
(5) It turns out that S. mansoni strains differ in virulence and pathology/disease outcomes (Jutzeler et al. 2024 - https://doi.org/10.1186/s13071-024-06286-6). Could the authors discuss how their results obtained from the Liberian strain are possibly relevant for other S. mansoni strains, e.g. the more virulent (and prevalent in the laboratories) PR strain?
Authors´ response: In a parallel study of one of us aiming at comparing the pathogenicity of different schistosome strains, evidence was obtained that the Puerto Rican (PR) strain showed higher virulence and advanced hepatic fibrosis compared to a Brasilian strain and the Liberian strain used in our study (Dannenhaus et al., in revision). Although we speculate that an inverse correlation between egg numbers and Th2 response might be found in the PR strain as well, perhaps even more pronounced, and in the Brasilian strain, it will be a matter of future studies to extend our approach to these and/or other schistosome strains to substantiate this hypothesis.
MINOR ISSUES
(a) The introduction should be divided into paragraphs for better orientation.
Authors´ response: The introduction has been divided into three sections in the revised manuscript. These address epidemiology, clinical picture and immunophenomena.
(b) Lines 53-54: "In this context, the liver plays an important role in evading host immunity as it is able to induce immune tolerance [12]." – The sentence seems a bit general and out of context.
Authors´ response: We agree with the reviewer. This sentence has been moved to the immunophenomena of the introduction.
(c) Lines 61-62: "In addition, co-infections with the human immunodeficiency virus (HIV), hepatitis B virus (HBV) or hepatitis C virus (HCV) are common [16,17]." – The sentence seems a bit out of context.
Authors´ response: We thank the reviewer for this remark. The sentence has been eliminated from the revised manuscript.
(d) Lines 72-73: I would specify the "immune response" as the authors analyzed only the local expression of certain cytokines.
Authors´ response: We agree with the reviewer. The term "immune response" was exchanged by the term “local cytokine expression”.
(e) Lines 74-75: I suggest rephrasing the reason for choosing IPSE/alpha-1, kappa-5, and omega-1. Although antibodies target them, this is not much relevant to the cytokine-oriented study. I assume the immunomodulatory capacity of those molecules plays the main role.
Authors´ response: We thank the reviewer for this remark and changed the sentence at the end of the introduction accordingly.
(f) Line 80: Please, indicate the infection dose.
Authors´ response: As indicated in the material and methods section hamsters were infected with S. mansoni cercariae by the paddling method [23] at the age of eight weeks. The golden hamsters were infected at the age of eight weeks by bath infection with cercariae in water at room temperature (2x 45 minutes, 1750 cercariae). The last sentence has been added to the revised manuscript.
(g) Line 81: It is unclear if the monosex (ms) infection was performed with male or female cercariae.
Authors´ response: We are happy to explain the somewhat cumbersome formulation. The monosex infection was carried out with female and with male cercariae. The cercariae were initially isolated and then multiplied, so that infection with both sexes but only with one sex per hamster occurred. To avoid misunderstandings the following sentence was added to the revised manuscript: “For the ms group cercariae were initially isolated and then multiplied, so that infection with both sexes but only with one sex per hamster occurred.”
(h) Line 86: How many non-infected (ni) animals were used?
Authors´ response: Six non-infected (ni) animals were used as super-controls. The number of super controls (ni) has been added to the amended manuscript.
(i) Line 84: It should be indicated that the hamsters were perfused (which I expect as the worm burden was also assessed in Fig. S1D).
Authors´ response: Indeed, the livers of the animals were perfused in situ, removed, shock-frozen with liquid nitrogen and stored at -80°C. This was added to the revised manuscript.
(j) Lines 92-95: It is unclear what kind of histochemical staining/detection was performed. Please, specify. It seems that only HE (no histochemistry) was used.
Authors´ response: As indicated in the revised section 2.2. Haematoxylin and eosin staining, sections of liver lobes from bisex-infected hamsters were stained with haematoxylin and eosin (H&E) in total cross-section.
(k) Line 112: Please, specify which part of the liver and colon was used for the analysis. Was the colon washed from the feces prior to weighting?
The requested issues were introduced appropriately: ´Schistosome eggs were isolated by digestion of randomly chosen parts of liver liver and colon tissue (without faeces) of bs-infected hamsters with potassium hydroxide (KOH) according to Cheever [30]. The even distribution of liver eggs is shown in figure 1.´
(l) Lines 117-123: I miss information on the regression/correlation analyses, which seem critical for most outcomes.
Authors´ response: The statistical correlation between various measurement parameters was analyzed using regression analyses. For this purpose, the measured variables were divided into dependent and independent parameters depending on their meaningfulness. The best regression function (linear, inverse or exponential) was determined using the SPSS® curve fitting function in order to mathematically describe the dependency of the parameters with as little deviation as possible. The maximum coefficient of determination R2 was used as a criterion. This indicates how much scatter in the data can be explained by the functional relationship.
(m) Figure 5: I would invert the axis, putting the molecules on X axis ("cytokine expression is dependant on the molecule expression").
Authors´ response: We would like to thank the reviewer for his valuable advice. Figure 5 has been amended accordingly.
(n) Lines 282-285: The sentence is very long and difficult to understand.
Authors´ response: lines 282-285 include two sentences: An infection with S. mansoni initially triggers a Th1 immune response followed by a shift to a Th2 response after the parasite starts to produce eggs [33]. Additionally, metabolic reprogramming and the development of reactive oxygen species (ROS) occur in the hepatic tissue [34,35].
The section has been rearranged slightly to make it easier to understand: An infection with S. mansoni initially triggers a Th1 immune response followed by a shift to a Th2 response shortly after egg production [33]. Additionally, egg production induces metabolic reprogramming and the development of reactive oxygen species (ROS) in hepatic tissue [34,35].
(o) Lines 274-300: This part is quite long but does not discuss anything. I believe it is largely a summary that could be condensed so that the reader can focus on the discussion itself.
Authors´ response: We agree. The first part of the discussion has been shortened significantly in the revised manuscript.
(p) Lines 383-384: "In this study, individual enzymes, cytokines, and transcription factors were evaluated as representative molecules." – I do not understand, which enzymes and transcription factors are addressed.
Authors´ response: We apologize for this inconsistency and have amended the section accordingly.
“In this study, individual cytokines representing Th1 and Th2 response, were evaluated as representative molecules.”
Reviewer 2 Report
Comments and Suggestions for Authors
- Line 81. The authors should clarify whether the monosex populations were all female worms, or some populations were male only.
- Line 101. Amend spelling of ‘Intron-spanning’.
- Line 137. Perhaps for clarity, say: ‘…and granulomas delineated by (---)’.
- Line 152. Replace ‘With’ by ‘Comprising’.
- Line 163. The reference to Fig. 1D seems superfluous here.
- Fig 1C. On the Y-axes, the authors should state the variable (example: relative expression level, as described in the legend).
- Suppl Fig 1C. For completeness, could the authors state or add to the graph the R-squared value?
- Figure 2A-D. On the Y-axes, the authors should state the variable (example: fold increase in expression compared to ni, as described in the legend).
- Fig 2C. The Y-axis upper limit is ‘2’, perhaps it should be ‘12’?
- Line 190. Should it be ‘…variable and fold increase in Ifng mRNA as the dependent variable’?
- Suppl Fig2. i) The authors could consider incorporating the current SFig S2 into the main text, as the contrast with liver is notable also for the significant differences in ms versus bs.
- Suppl Fig2. ii) Amend spelling of ‘induced’ in Legend part (A).
- Figure 3A-F. On the Y-axes, the authors should state the variable (example: fold increase in expression compared to ms- or ni-groups, as described in the legend).
- Line 214/215. Amend to ‘(Fig. 4A)’ and ‘(Fig. 4B)’ .
- Line 220. Amend to ‘(A, C, and D)’.
- Figure 5A-F. On the X- and Y-axes, the authors should state the variable (example: fold increase in expression compared to ms-group).
- Line 249. The abbreviation AAM should be explained here at its first use.
- Figure 6A-D. On the X- and Y-axes, the authors should state the variable (example: fold increase in expression compared to ni-group).
- Line 360. Amend spelling of ‘lumen’
Author Response
Answers to reviewer 2
We would like to thank Reviewer 2 for his consistently positive assessment of our work.
- Line 81. The authors should clarify whether the monosex populations were all female worms, or some populations were male only.
Authors´ response: We would like to explain the somewhat cumbersome formulation as mentioned above. The monosex infection was carried out with female and with male cercariae. The cercariae were initially isolated and then multiplied, so that infection with both sexes but only with one sex per hamster occurred. To avoid misunderstandings the following sentence was added to the revised manuscript: “For the ms group cercariae were initially isolated and then multiplied, so that infection with both sexes but only with one sex per hamster occurred.”
- Line 101. Amend spelling of ‘Intron-spanning’.
Authors´ response: The word intron-spanning was corrected.
- Line 137. Perhaps for clarity, say: ‘…and granulomas delineated by (---)’.
Authors´ response: We improved the revised manuscript accordingly.
- Line 152. Replace ‘With’ by ‘Comprising’.
Authors´ response: We improved the revised manuscript accordingly.
- Line 163. The reference to Fig. 1D seems superfluous here.
Authors´ response: We improved the revised manuscript accordingly and removed the reference to Figure 1D.
- Fig 1C. On the Y-axes, the authors should state the variable (example: relative expression level, as described in the legend).
Authors´ response: We improved all figures accordingly.
- Suppl Fig 1C. For completeness, could the authors state or add to the graph the R-squared value?
Author´ response: Thank you for this advice. We completed suppl Fig1C accordingly.
- Figure 2A-D. On the Y-axes, the authors should state the variable (example: fold increase in expression compared to ni, as described in the legend).
Authors´ response: We improved the figure 2A-D. (2A-D). In regard to the available space in
- Fig 2C. The Y-axis upper limit is ‘2’, perhaps it should be ‘12’?
Authors´ response: We apologize for this mistake and corrected the number in figure 2C.
- Line 190. Should it be ‘…variable and fold increase in Ifng mRNA as the dependent variable’?
Authors´ response: We thank the reviewer for this remark and changed the legend of figure 2 accordingly.
- Suppl Fig2. i) The authors could consider incorporating the current SFig S2 into the main text, as the contrast with liver is notable also for the significant differences in ms versus bs.
Authors´ response: We thank the reviewer for this advice and incorporated the SFig S2 into the main and revised manuscript as Fig. 2E-H.
- Suppl Fig2. ii) Amend spelling of ‘induced’ in Legend part (A).
Author´ response: We improved the figure legend accordingly.
- Figure 3A-F. On the Y-axes, the authors should state the variable (example: fold increase in expression compared to ms- or ni-groups, as described in the legend).
Authors´ response: We improved the figure 3A-F accordingly. In regard to the available space in the figure, we mention the comparison in the legend only.
- Line 214/215. Amend to ‘(Fig. 4A)’ and ‘(Fig. 4B)’ .
Authors´ response: We improved the revised manuscript accordingly.
- Line 220. Amend to ‘(A, C, and D)’.
Authors´ response: We improved the revised manuscript accordingly
- Figure 5A-F. On the X- and Y-axes, the authors should state the variable (example: fold increase in expression compared to ms-group).
Authors´ response: We would like to thank the reviewer for his valuable advice. Figure 5 has been amended accordingly. In regard to the available space in the figure, we mention the comparison in the legend only.
- Line 249. The abbreviation AAM should be explained here at its first use.
Authors´ response: We improved the revised manuscript accordingly and explained AAM at its first use.
- Figure 6A-D. On the X- and Y-axes, the authors should state the variable (example: fold increase in expression compared to ni-group).
Authors´ response: We improved the figure 6A-6D accordingly. In regard to the available space in the figure, we mention the comparison in the legend only.
- Line 360. Amend spelling of ‘lumen’:
Authors´ response: lumen has been corrected.
Thanks to the reviewer. We improved the revised manuscript accordingly.

Reviewer 3 Report
Comments and Suggestions for Authors
Thank you for sharing your article on inverse correlation of Th2-specific cytokines with hepatic egg burden in hamsters infected with S. mansoni. The following comments may help to improve the article:
L80: How many cercariae received each hamster? How many hamsters were infected, and were they males or females or mixed? Please revise accordingly.
L87: How many control animals did you use? Please revise accordingly.
L101-110: Consider listing primers in a separate table also for better readability.
Graphical abstract: Please add units and scaling to the x- and z-axes.
Figure 1 and Figures 2-6: Please use the same max. scaling for both x-axis and z-axis for better readability and comparability of the figures.
Comments on the Quality of English Language
Especially the introduction reads a bit "rocky" and benefit for professional English editing.
Author Response
Answers to reviewer 3
We thank Reviewer 3 for the kind words about our manuscript.
Major issues:
L80: How many cercariae received each hamster? How many hamsters were infected, and were they males or females or mixed? Please revise accordingly.
Authors´ response: We added or modified the following sentences to the materials and methods part to clear these points (line no. refers to the revised manuscript with changes accepted):
Line 78: ´Biomphalaria glabrata snails served as intermediate hosts and female Syrian hamsters (Mesocricetus auratus) as final hosts for maintaining the life-cycle of a Liberian strain of S. mansoni [22].´
Line 80: ´The golden hamsters were infected at the age of eight weeks by bath infection with cercariae in water at room temperature (45 minutes, 1750 cercariae).´
Line 86: ´In total, 22 bs-infections were carried out at 8 weeks of (hamster) age, and infected animals maintained for 46 days. In addition, 10 ms-infections were carried out for 67 days to ensure a complete maturation of the worms [24]; …´
L87: How many control animals did you use? Please revise accordingly.
Authors´ response: The no. of control animals was introduced in line 88 of the revised manuscript with changes accepted: ´Six non-infected (ni) animals were used as super-controls.´
L101-110: Consider listing primers in a separate table also for better readability.
Authors´ response: We moved the primers to Supplementary Table S2: Primer list for better readability.
Graphical abstract: Please add units and scaling to the x- and z-axes.
Authors´ response: The graphical abstract should summarize the major findings and conclusions schematically in an abstracted form to transfer the summary on a glance without details. Therefore, we considered not to scale the axes.
Figure 1 and Figures 2-6: Please use the same max. scaling for both x-axis and z-axis for better readability and comparability of the figures.
Authors´ response: The scaling of axes was optimized in the revised manuscript. Please keep in mind that the relative expression levels of distinct genes are not directly comparable for liver and colon as exemplarily mentioned for figures 1 C and D in the text (line 195).
Round 2
Reviewer 1 Report
Comments and Suggestions for Authors
I was happy to read the revised version of the manuscript. I sincerely appreciate all the changes made by the authors. They addressed all my issues. I have just two remaining minor points which remain to be clarified:
(1) Do I understand correctly that the ms group was infected by either 1750 male or female cercariae? And then the male- of female-infected hamsters were pooled together to get n=10? If so, it seems quite interesting to me that the ms groups had similar or even lower variation than the bs group. In some parameters, I would expect higher variation or even bimodal distribution, having in mind likely differential effects of the sexes on the host (e.g., https://doi.org/10.1371/journal.pntd.0005595, https://doi.org/10.3389/fimmu.2018.00861, https://doi.org/10.3389/fcimb.2022.893632, https://doi.org/10.3389/fimmu.2022.1010932).
(2) Peterkova et al. [50] are wrongly cited in the reference list (McKenzie et al. are shown in the reference list).
Author Response
Answers to Reviewer 1
I was happy to read the revised version of the manuscript. I sincerely appreciate all the changes made by the authors. They addressed all my issues. I have just two remaining minor points which remain to be clarified:
Authors´ response: We thank Reviewer 1 for the kind words about of our manuscript.
(1) Do I understand correctly that the ms group was infected by either 1750 male or female cercariae? And then the male- of female-infected hamsters were pooled together to get n=10? If so, it seems quite interesting to me that the ms groups had similar or even lower variation than the bs group. In some parameters, I would expect higher variation or even bimodal distribution, having in mind likely differential effects of the sexes on the host (e.g., https://doi.org/10.1371/journal.pntd.0005595, https://doi.org/10.3389/fimmu.2018.00861, https://doi.org/10.3389/fcimb.2022.893632, https://doi.org/10.3389/fimmu.2022.1010932).
Authors´ response: Yes, the reviewer understood it perfectly. This is a very interesting point and we agree with the reviewer that the differential effects of the parasite´s sexes on the host might enhance the variation in some parameters. Nevertheless, this was not the focus of our work and we have focused on the pathologies caused by the eggs. We assume that the stronger hepatic pathology in bs-infected hamsters and the variation in hepatic egg load are responsible for the higher variations of the analyzed parameters in bs-infected animals in comparison to the ms-controls.
(2) Peterkova et al. [50] are wrongly cited in the reference list (McKenzie et al. are shown in the reference list).
Authors´ response: We apologize for this inconsistency and corrected the reference list accordingly.
Reviewer 2 Report
Comments and Suggestions for Authors
I thank the authors for the clarifications. Congratulations on an interesting manuscript.
Author Response
Answers to Reviewer 2
I thank the authors for the clarifications. Congratulations on an interesting manuscript.
Authors´ response: We thank Reviewer 2 for the kind words about of our manuscript.
Reviewer 3 Report
Comments and Suggestions for Authors
Thank you for addressing most of my comments and suggested edits. Optimising the scaling of the axes of figures has improved their readability and comparability despite presented liver or colon data certainly cannot be compared directly. From an objective reader's point of view the figures' axes could be optimised more.
Author Response
Answers to Reviewer 3
Thank you for addressing most of my comments and suggested edits. Optimising the scaling of the axes of figures has improved their readability and comparability despite presented liver or colon data certainly cannot be compared directly. From an objective reader's point of view the figures' axes could be optimised more.
Authors´ response: We thank Reviewer 3 for the valuable suggestions to improve our manuscript.